# Training Physics-Driven Deep Learning Reconstruction without Raw Data Access for Equitable Fast MRI

## Abstract

Physics-driven deep learning (PD-DL) approaches have become popular for improved reconstruction of fast magnetic resonance imaging (MRI) scans. Even though PD-DL offers higher acceleration rates compared to existing clinical fast MRI techniques, their use has been limited outside specialized MRI centers. One impediment for their deployment is the difficulties with generalization to pathologies or population groups that are not well-represented in training sets. This has been noted in several studies, and fine-tuning on target populations to improve reconstruction has been suggested. However, current approaches for PD-DL training require access to raw k-space measurements, which is typically only available at specialized MRI centers that have research agreements for such data access. This is especially an issue for rural and underserved areas, where commercial MRI scanners only provide access to a final reconstructed image. To tackle these challenges, we propose **C**ompressibility-inspired **U**nsupervised Learning via **P**arallel **I**maging Fi**d**elity (CUPID) for high-quality PD-DL training using only routine clinical reconstructed images exported from an MRI scanner. CUPID evaluates the goodness of the output with a compressibility-based approach, while ensuring that the output stays consistent with the clinical parallel imaging reconstruction through well-designed perturbations. Our results show that CUPID achieves similar quality compared to well-established PD-DL training strategies that require raw k-space data access, while outperforming conventional compressed sensing (CS) and state-of-the-art generative methods. We also demonstrate its effectiveness in a zero-shot training setup for retrospectively and prospectively subsampled acquisitions, attesting to its minimal training burden. As an approach that radically deviates from existing strategies, CUPID presents an opportunity to provide equitable access to fast MRI for underserved populations in an attempt to reduce the inequalities associated with this expensive imaging modality.

## 1 Introduction

Magnetic resonance imaging (MRI) is a central tool in modern medicine, offering multiple soft tissue contrasts and high diagnostic sensitivity for numerous diseases. However, MRI is among the most expensive medical imaging modalities, in part due to its long scan times. Demand for MRI scans has shown an annual growth rate of 2.5%, while the number of MRI units per capita has increased by 1.8% in a similar time frame (Martella et al., 2023). This mismatch has further increased the wait times for MRI exams (Bartsch et al., 2024; Hofmann et al., 2023), particularly in rural areas and underserved communities (Burdorf, 2022), as depicted in Fig. 1. Thus, techniques for fast MRI scanning that can reduce overall scan times without compromising diagnostic quality (Akçakaya et al., 2022) are critical for improving the throughput of MRI.

Computational MRI approaches, including partial Fourier imaging (McGibney et al., 1993), parallel imaging (Pruessmann et al., 1999; Griswold et al., 2002), compressed sensing (Lustig et al., 2007), and more recently deep learning (DL) (Hammernik et al., 2018; Schlemper et al., 2018) have been developed for accelerating MRI. In most MRI centers, parallel imaging remains the most widely used approach for the reconstruction of routine clinical images. The acceleration rates afforded by these methods, however, are limited due to noise amplification and aliasing artifacts. DL-based

**a) Access to MRI in Minnesota by County**

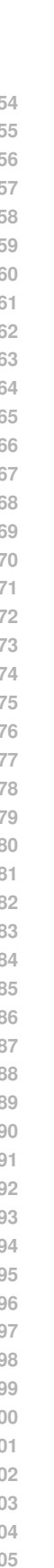

In-hospital MRI

Mobile MRI Only

No MRI Access

**b) MRI scanners outside specialized centers**

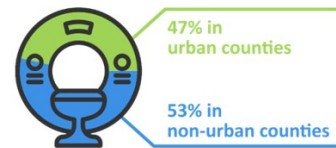

47% in urban counties

53% in non-urban counties

- Non-urban MRI scanners do not enjoy the specialized expertise in academic medical centers
- No vendor research agreements, needed for raw data access
- Cannot use or fine-tune fast MRI methods that require raw data

Figure 1: Many areas in the world have limited access to MRI or rely on services from local hospitals and mobile MRI: (a) A recent study from Minnesota, United States (Burdorf, 2022) show more than half of MRI services are in non-urban counties. (b) These non-urban MRI machines are unlikely to have vendor agreements that provide raw MRI data access, precluding access to and fine-tuning of AI-based fast MRI techniques.

methods, especially physics-driven DL (PD-DL) approaches, offer state-of-the-art improvements over parallel imaging (Knoll et al., 2020a). However, the translation of PD-DL to clinic has been hindered by generalizability and artifact issues related to details not well represented in training databases, in other words when faced with out-of-distribution samples at test time (Eldar et al., 2017; Knoll et al., 2020b; Muckley et al., 2021; Antun et al., 2020). This is a problem for many typical MRI centers, whose population characteristics do not necessarily align with specialized MRI centers in urban settings, where training databases for PD-DL are currently curated.

In such cases, fine-tuning of the PD-DL model on the target population may be beneficial (Knoll et al., 2019; Dar et al., 2020; Yaman et al., 2022b; Chandler et al., 2023). However, a major road-block for this strategy is that all current training methods for PD-DL require access to raw MRI data. Such access requires research agreements with MR vendors, and is typically not available outside specialized/academic MRI centers. This is especially an issue for rural and underserved areas, where commercial MRI scanners only provide access to a final image, reconstructed via parallel imaging.

In this work, we tackle these challenges associated with typical PD-DL training, and propose **C**ompressibility-inspired **U**nsupervised Learning via **P**arallel **I**maging Fi**d**elity (CUPID), which trains PD-DL reconstruction from routine clinical images, for instance in Digital Imaging and Communications in Medicine (DICOM) format. Succinctly, CUPID uses a compressibility-inspired term to evaluate the goodness of the output, while ensuring the output is consistent with parallel imaging via well-designed input perturbations. CUPID can be used both with database-training and in a subject-specific/zero-shot manner, attesting to its minimal fine-tuning burden.

Our key contributions include:

- We introduce CUPID a novel method that enables high-quality training of PD-DL reconstruction in unsupervised and zero-shot/subject-specific settings using only routine clinical reconstructed MRI images, eliminating the need for access to raw k-space measurements. To the best of our knowledge, our method is the first attempt to train PD-DL networks using only these images that are exported from the scanner.

- CUPID trains on DICOM images acquired at the *target acceleration rate*, which often have noise and aliasing artifacts due to high sub-sampling, in an unsupervised manner. Note this is a deviation from other methods that use reference fully-sampled DICOM images to train a likelihood or score function, such as generative models.

- CUPID uses a novel unsupervised loss formulation that enforces fidelity with using parallel imaging algorithms via carefully designed perturbations, in addition to evaluating the compressibility of the output image. This parallel imaging fidelity ensures the network does not converge to overly sparse solutions.

- We provide a comprehensive evaluation, encompassing acquisitions with both retrospective and prospective undersampling at target acceleration rates, to demonstrate that CUPID achieves results on par with leading supervised and self-supervised training strategies that depend on raw k-space data, while surpassing conventional compressed sensing (CS) techniques and state-of-the-art generative methods.
- CUPID enables training/fine-tuning of PD-DL reconstruction from routine clinically reconstructed DICOM images, and may be instrumental to provide equitable access to fast MRI methods, especially in underserved and rural areas.

# 2 BACKGROUND AND RELATED WORK

## 2.1 MRI FORWARD MODEL AND CONVENTIONAL METHODS FOR MRI RECONSTRUCTION

MRI raw data is acquired in the frequency domain of the image, referred to as k-space. For fast MRI, data is acquired in the sub-Nyquist regime by undersampling the acquisition in k-space. In this case, the forward acquisition model relating the image $\mathbf{x} \in \mathbb{C}^{\mathbf{n}}$ to these raw MRI data (or k-space) measurements is given as:

$$\mathbf{y}_\Omega = \mathbf{E}_\Omega \mathbf{x} + \mathbf{n}, \tag{1}$$

where $\mathbf{y}_\Omega$ denotes the acquired k-space data corresponding to the undersampling pattern $\Omega$ with $|\Omega| = m < n$. $\mathbf{E}_\Omega$ denotes the multi-coil encoding operator that includes information from $n_c$ receiver coils, each of which are sensitive to a different part of the image (Hamilton et al., 2017). When the acceleration rate $R = n/m$ is less than $n_c$, this system of equations is over-determined due to the redundancies among the receiver coils. Parallel imaging uses these redundancies to solve the maximum likelihood estimation problem under i.i.d. Gaussian noise (Pruessmann et al., 1999):

$$\mathbf{x}_{\text{PI}} = \arg\min_{\mathbf{x}} \|\mathbf{y}_\Omega - \mathbf{E}_\Omega \mathbf{x}\|_2^2 = (\mathbf{E}_\Omega^H \mathbf{E}_\Omega)^{-1} \mathbf{E}_\Omega^H \mathbf{y}_\Omega. \tag{2}$$

Numerically, this can be solved directly for certain undersampling patterns (Pruessmann et al., 1999) or more broadly iteratively using conjugate gradient (CG) (Pruessmann et al., 2001). Using the equivalence of multiplication in image domain and convolutions in k-space (Uecker et al., 2014), it can also be solved as an interpolation problem in k-space (Griswold et al., 2002). Parallel imaging remains the most clinically used acceleration method for MRI, with some MR systems using the image-based reconstruction, while others utilizing the equivalent k-space interpolation.

In modern computational MRI, additional regularization is often incorporated into the objective function (Hammernik et al., 2023):

$$\arg\min_{\mathbf{x}} \|\mathbf{y}_\Omega - \mathbf{E}_\Omega \mathbf{x}\|_2^2 + \mathcal{R}(\mathbf{x}), \tag{3}$$

where $\mathcal{R}(\cdot)$ denotes a regularizer. For instance, compressed sensing (CS) uses the idea that images should be compressible in an appropriate transform domain (Lustig et al., 2007), and uses $\mathcal{R}(\mathbf{x}) = \tau \|\mathbf{W}\mathbf{x}\|_1$, where $\tau$ is the regularization weight, $\mathbf{W}$ is a linear sparsifying transform such as a discrete wavelet transform (DWT) and $\|\cdot\|_1$ is the $\ell_1$ norm.

## 2.2 PD-DL RECONSTRUCTION VIA ALGORITHM UNROLLING

Among different PD-DL methods (Ahmad et al., 2020; Gilton et al., 2021; Knoll et al., 2020a), unrolled networks (Monga et al., 2021) remain the highest performer in reconstruction challenges, as reported a year ago (Hammernik et al., 2023; Muckley et al., 2021). These methods unroll iterative algorithms for solving the regularized least squares objective in (3) (Fessler, 2020), such as proximal gradient descent (Schlemper et al., 2018) or variable splitting with quadratic penalty (VS-QP) (Aggarwal et al., 2019), over a fixed number of steps. VS-QP transforms (3) into 2 sub-problems:

$$\mathbf{z}^{(i)} = \arg\min_{\mathbf{z}} \|\mathbf{x}^{(i-1)} - \mathbf{z}\|_2^2 + \mathcal{R}(\mathbf{z}), \tag{4a}$$

$$\mathbf{x}^{(i)} = \arg\min_{\mathbf{x}} \|\mathbf{y}_\Omega - \mathbf{E}_\Omega \mathbf{x}\|_2^2 + \mu \|\mathbf{x} - \mathbf{z}^{(i)}\|_2^2, \tag{4b}$$

where (4a) is the proximal operator for the regularization, implicitly solved using neural networks, while (4b) accounts for data fidelity and has a closed form solution:

$$\mathbf{x}^{(i)} = \left(\mathbf{E}_\Omega^H \mathbf{E}_\Omega + \mu \mathbf{I}\right)^{-1} \left(\mathbf{E}_\Omega^H \mathbf{y}_\Omega + \mu \mathbf{z}^{(i)}\right), \tag{5}$$

which can be solved by CG (Aggarwal et al., 2019). Unrolled networks are conventionally trained using supervised learning over a database, where the reference raw k-space measurements are first retrospectively undersampled to form $\mathbf{y}_\Omega$. Subsequently, the network is trained to map to the original full reference k-space or the corresponding reference image (Hammernik et al., 2018; Aggarwal et al., 2019) by minimizing:

$$\min_{\boldsymbol{\theta}} \mathbb{E}\left[\mathcal{L}\left(\mathbf{y}_{\text{ref}}, \mathbf{E}_{\text{full}}(f(\mathbf{y}_\Omega, \mathbf{E}_\Omega; \boldsymbol{\theta}))\right)\right] \tag{6}$$

where $\boldsymbol{\theta}$ are the network parameters, $f(\mathbf{y}_\Omega, \mathbf{E}_\Omega; \boldsymbol{\theta})$ denotes the network output for inputs $\mathbf{y}_\Omega$ and $\mathbf{E}_\Omega$, $\mathbf{E}_{\text{full}}$ is the fully-sampled encoding operator, $\mathbf{y}_{\text{ref}}$ is the fully-sampled reference k-space data, and $\mathcal{L}(\cdot, \cdot)$ is a loss function.

## 2.3 SELF-SUPERVISED AND UNSUPERVISED METHODS

Obtaining fully-sampled reference data in MRI can be infeasible due to prolonged scan durations, organ movement in acquisitions such as real-time cardiac imaging or myocardial perfusion (Rajiah et al., 2023), or signal decay in acquisitions like diffusion MRI with EPI (Uğurbil et al., 2013). To enable training of PD-DL networks without fully sampled raw MRI data, a variety of unsupervised learning methodologies have emerged (Akçakaya et al., 2022), including self-supervised learning techniques (Yaman et al., 2020; Chen et al., 2021) and generative modeling approaches (Jalal et al., 2021; Chung & Ye, 2022; Chung et al., 2023).

Self-supervised methods use a masking approach to generate supervisory labels from the undersampled data (Yaman et al., 2020; Millard & Chiew, 2023; Hu et al., 2024). A pioneering method in this field, self-supervision via data undersampling (SSDU) (Yaman et al., 2020; 2022a), involves partitioning the acquired measurement $\Omega$ into two disjoint subsets ($\Omega = \Lambda \cup \Theta$) to train the network in a self-supervised manner:

$$\min_{\boldsymbol{\theta}} \mathbb{E}\left[\mathcal{L}\left(\mathbf{y}_\Lambda, \mathbf{E}_\Lambda(f(\mathbf{y}_\Theta, \mathbf{E}_\Theta; \boldsymbol{\theta}))\right)\right] \tag{7}$$

Even though these self-supervision based approaches demonstrate exceptional performance across various tasks, they lack the ability to train the model without access to undersampled raw data, as they cannot operate solely using images that are exported from the scanner.

Conversely, generative methods learn the prior distribution of the given dataset, which is then leveraged in conjunction with a log-likelihood data term during the testing phase. Although recent methods based on diffusion/score-based models have shown substantial promise, these methods require large amounts of high-quality images either reconstructed from raw data (Jalal et al., 2021; Luo et al., 2023) or as DICOMs (Chung & Ye, 2022), as well as computational resources to perform the training, both of which may not be feasible in the setups we are focused on.

## 3 UNSUPERVISED TRAINING FOR PD-DL WITHOUT RAW K-SPACE DATA

In this study, we introduce a novel framework to train PD-DL models, utilizing only routinely available clinical images exported directly from MRI scanners. Recently, inspired by the connections between PD-DL and compressibility-based processing (Gu et al., 2022), a compressibility-inspired loss was proposed to evaluate the goodness of unsupervised PD-DL training (Alçalar et al., 2024). However, this approach still requires access to raw k-space data to stabilize training, making it unsuitable for our goals. Here, we adapt the compressibility idea and augment it with a parallel imaging fidelity to successfully reconstruct clinical images in DICOM format without needing any raw k-space data.

**Compressibility Aspect of the Loss Formulation.** Compressibility/sparsity in the output of the PD-DL network can be enforced by utilizing a weighted $\ell_1$ norm (Alçalar et al., 2024), which has been demonstrated to provide a closer approximation to the $\ell_0$ norm compared to the standard $\ell_1$ norm (Candes et al., 2008). Thus, this compressibility of the output image in CUPID is achieved by the loss term

$$\mathcal{L}_{\text{comp}}(\mathbf{x}_{\text{PI}}, \mathbf{m}) = \frac{1}{N} \cdot \sum_{n=1}^{N} \left( \frac{|(\mathbf{W}f(\mathbf{x}_{\text{PI}}, \mathbf{E}_\Omega))_n|}{|(\mathbf{W}\mathbf{x}^{(\text{m})})_n| + \epsilon} \right), \tag{8}$$

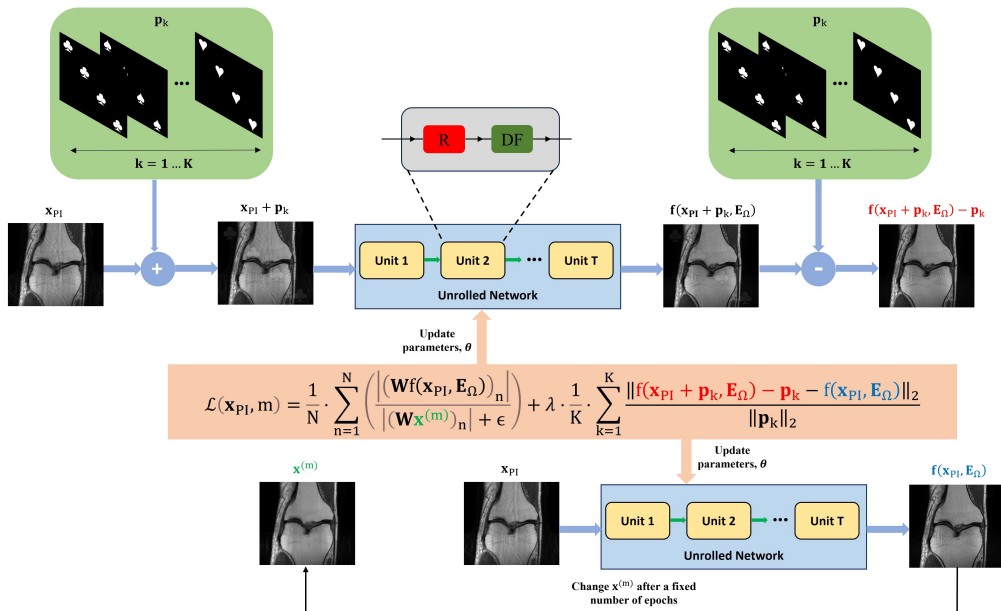

Figure 2: Our Compressibility-inspired Unsupervised Learning via Parallel Imaging Fidelity (CU-PID) method trains PD-DL models in an unsupervised and/or zero-shot manner without requiring any raw k-space data. The network is unrolled for $T$ units, with each unit consisting of regularizer (R) and data fidelity (DF). The first term in the loss function is a reweighted $\ell_1$ component designed to assess the compressibility of the network's output, while the second term is a fidelity term that ensures the network does not produce a zero output by carefully perturbing the input data to check if it stays consistent with the parallel imaging reconstruction.

where $\mathbf{x}_{\mathrm{PI}}$ denotes the DICOM input acquired using parallel imaging, $\mathbf{W}$ represents the wavelet transform, $N$ is the total number of wavelet coefficients and $\mathbf{x}^{(m)}$ signifies the signal estimate following the training during the $m^{\mathrm{th}}$ reweighting step. Similar to Alçalar et al. (2024); Candes et al. (2008), we chose the initial weights from a CS reconstruction that has a large regularization and $\epsilon$ is added for numerical stability. Note, here we redefined $f(\cdot, \cdot)$ without the network parameters, $\boldsymbol{\theta}$, and used $\mathbf{x}_{\mathrm{PI}}$ as the network input instead of $\mathbf{y}_{\boldsymbol{\Omega}}$, to simplify notation.

**Parallel Imaging Fidelity.** Relying solely on (8) will result in inaccurate training as the neural network learns to produce an all-zeros image in an effort to drive the wavelet coefficients in the numerator to zero, which minimizes the loss function in (8). In Alçalar et al. (2024), fidelity with raw k-space data was used to avoid this training issue. In our setting without raw k-space access, we introduce a novel fidelity operator that stabilizes the training of the reconstruction algorithm, building on ideas from parallel imaging.

Specifically, we ensure that our network outputs are consistent with any clinical parallel imaging reconstruction through carefully crafted perturbations, $\{\mathbf{p}_k\}$. These perturbations for $R$-fold acceleration are designed in such a way that $R$-fold aliasing do not create overlaps in the field-of-view, indicating that they could be resolved by parallel imaging reconstruction. The idea behind this design choice is to ensure that the network, when applied to the unperturbed $\mathbf{x}_{\mathrm{PI}}$, yields an accurate estimate of $\mathbf{x}$, and when applied to $\mathbf{x}_{\mathrm{PI}} + \mathbf{p}$, similarly recovers $\mathbf{x} + \mathbf{p}$, as the perturbation $\mathbf{p}$ must be resolvable within the framework of any parallel imaging approach. Both processes are visualized in Fig. 2. By doing so, the consistency term ensures a non-zero output when the sparsity is minimized. Thus, our second loss term that enforces parallel imaging fidelity is given as:

$$\mathcal{L}_{\mathrm{pif}}(\mathbf{x}_{\mathrm{PI}}) = \mathbb{E}_{\mathbf{p}}\left[\frac{||f(\mathbf{x}_{\mathrm{PI}} + \mathbf{p}, \mathbf{E}_{\boldsymbol{\Omega}}) - \mathbf{p} - f(\mathbf{x}_{\mathrm{PI}}, \mathbf{E}_{\boldsymbol{\Omega}})||_2}{||\mathbf{p}||_2}\right]. \tag{9}$$

From an implementation perspective, the expectation over $\mathbf{p}$ is calculated over $K$ such perturbations $\{\mathbf{p}_k\}$. The fold-over constraint for each $\{\mathbf{p}_k\}$ is achieved by picking the perturbations as randomly rotated and positioned letters, numbers, card suits or other shapes that have different intensity

values. These choices also ensure that high-frequency information, such as edges, are accurately reconstructed by the regularization process. Our final loss function for CUPID is:

$$\mathcal{L}_{\text{CUPID}} = \mathcal{L}_{\text{comp}} + \lambda \cdot \mathcal{L}_{\text{pif}}, \tag{10}$$

where $\lambda$ is a trade-off parameter between two terms.

**Subject-Specific / Zero-Shot Application.** In resource-limited or underserved settings, it may be more practical to fine-tune the method using only a few subjects, or even a single subject, to significantly reduce computational costs. As (8) does not solely focus on the subtraction between two entities, it lacks an inherent mechanism to drive the loss to zero through overfitting. Therefore, CUPID can be tailored to suit a scan-specific context (Akçakaya et al., 2019) without any modification to the loss given in (10).

## 4 EVALUATION

### 4.1 EXPERIMENTAL SETUP AND IMPLEMENTATION DETAILS

We conducted a thorough evaluation of our method, assessing its performance through both qualitative and quantitative analyses, and focused on uniform/equidistant patterns which produces coherent artifacts that are more difficult to remove compared to the incoherent artifacts from random undersampling (Knoll et al., 2019). We further note that CUPID demonstrates robust performance across a wide range of $\lambda$ values, provided that $\lambda$ is chosen within a reasonable range. An ablation study on the choice of $\lambda$ is included in Sec. 4.5.

**Retrospective Undersampling Setup.** In our retrospective studies, we used fully-sampled multi-coil knee and brain MRI data from the fastMRI database (Knoll et al., 2020b). Knee dataset included fully-sampled coronal proton density-weighted (coronal PD) and PD with fat suppression (coronal PD-FS) data. For brain MRI, axial FLAIR (ax-FLAIR) dataset with matrix size of $320 \times 320$ is used. The knee and brain MRI datasets comprised data collected from 15 and 20 receiver coils, respectively. Both datasets were retrospectively undersampled using a uniform/equidistant pattern at $R = 4$. 24 lines of auto-calibration signal (ACS) from center of the raw k-space data were kept. DICOM images to train our proposed model were reconstructed using parallel imaging (CG-SENSE), solving $\mathbf{x}_{\text{PI}} = (\mathbf{E}_\Omega^H \mathbf{E}_\Omega)^{-1} \mathbf{E}_\Omega^H \mathbf{y}_\Omega$. For each dataset, models were trained using 300 slices, and testing was performed using 380 slices for knee MRI and 100 slices for brain MRI, from distinct subjects.

**Prospective Undersampling Setup.** A multi-echo 3D GRE sequence on a 7T Siemens Magnetom MRI scanner was acquired. In this experiment, we replicate the practical pipeline for CUPID, where data is acquired at the desired high acceleration rate, and reconstructed to $\mathbf{x}_{\text{PI}}$ with noise and aliasing artifacts, using parallel imaging. The corresponding DICOM images are exported and used for fine-tuning the PD-DL model with CUPID. To this end, the brain dataset, with matrix size $= 288 \times 288$ and in-plane resolution $0.7 \times 0.7\text{mm}^2$, was acquired with prospective undersampling $R = 9$ (in $k_y$ only), which is the desired target acceleration, much higher than the clinical protocol at $R = 3$. Low-resolution images were acquired in the same orientation for sensitivity estimation (Krueger et al., 2023). Training and reconstruction with CUPID was done in a zero-shot subject-specific manner.

More details about the implementation of the PD-DL models are provided in Appendix A.

### 4.2 COMPARISON METHODS

We compared our method against several database training methods that have access to raw k-space data, including supervised PD-DL (Hammernik et al., 2018; Aggarwal et al., 2019; Knoll et al., 2020a), self-supervision via data undersampling (SSDU) (Yaman et al., 2020), and equivariant imaging (EI) (Chen et al., 2021). All PD-DL methods utilized the same unrolled network and components (Appendix A) to ensure that only the training process differed for fair comparisons.

In addition, we compared our approach with methods that can operate without raw data access as long as $\mathbf{E}_\Omega$ is known at test time. These include compressed sensing (CS) (Lustig et al., 2007), and ScoreMRI (Chung & Ye, 2022). The latter trains a time-dependent score function using denoising score matching on a large dataset of reference fully-sampled images, and uses this score function

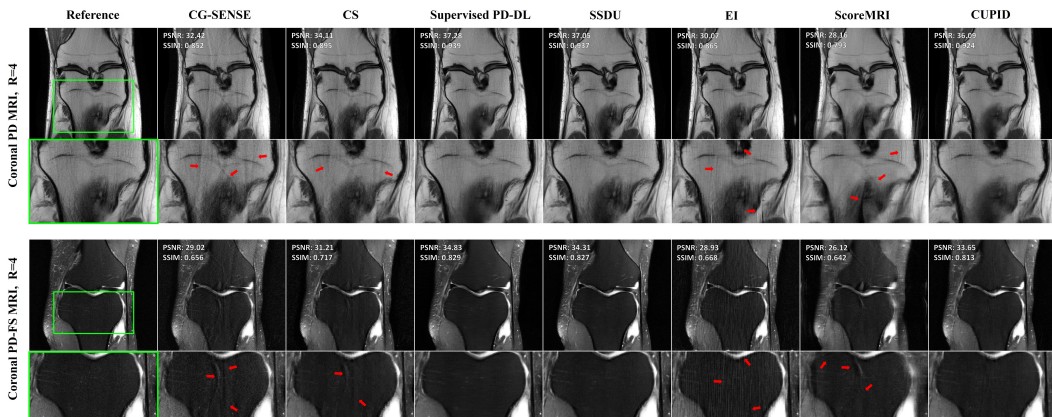

Figure 3: Representative coronal PD and PD-FS knee slices reconstructed using different methods for retrospective $R = 4$ uniform/equidistant undersampling. The baseline CG-SENSE, conventional CS, EI-trained PD-DL, and ScoreMRI suffers from residual artifacts highlighted by red arrows. PD-DL trained with CUPID loss function improves upon them while delivering reconstruction quality comparable to SSDU-trained PD-DL and supervised PD-DL.

during inference to sample from the conditional distribution given the measurements. Note both CS and ScoreMRI use $\mathbf{E}_{\Omega}^{H}\mathbf{y}_{\Omega}$ for data fidelity during inference. This can be accessed by multiplying $\mathbf{x}_{\text{PI}}$ with $\mathbf{E}_{\Omega}^{H}\mathbf{E}_{\Omega}$. Note that $\mathbf{E}_{\Omega}$ includes information about the undersampling pattern $\Omega$, which is completely known from the acquisition parameters, and coil sensitivities, which can be estimated from separate calibration scans in DICOM format (Krueger et al., 2023). A similar observation applies to the data fidelity in (5) for unrolled networks, thus they can be used for inference using only $\mathbf{x}_{\text{PI}}$ and $\mathbf{E}_{\Omega}$. We emphasize that what sets CUPID apart from other PD-DL strategies is that it is the only one that can train the unrolled network without using $\mathbf{y}_{\Omega}$. Thus, without loss of generality, $\mathbf{E}_{\Omega}$ is known both at training and testing for all methods. Finally, we also used CG-SENSE, which was used to generate the original $\mathbf{x}_{\text{PI}}$ as the clinical baseline comparison.

In the zero-shot setup, we compared our zero-shot results with zero-shot SSDU (ZS-SSDU) (Yaman et al., 2022b) as well as ScoreMRI, compressed sensing (CS), and our baseline method, CG-SENSE - all of which are compatible with zero-shot inference. All quantitative evaluations used structural similarity index (SSIM) and peak signal-to-noise ratio (PSNR).

## 4.3 EXPERIMENTS WITH RETROSPECTIVELY UNDERSAMPLED DATA

**Database Results.** Representative results in Fig. 3 show that baseline CG-SENSE, CS, EI and ScoreMRI reconstructions exhibit residual artifacts. In contrast, CUPID successfully eliminates

Table 1: Quantitative results for comparison methods on Coronal PD, Coronal PD-FS and Ax-FLAIR datasets using equispaced undersampling pattern at $R = 4$. Top 3 rows: Access to raw data (for training); Last 4 rows: No raw data access.

| Method | Coronal PD | | Coronal PD-FS | | Ax-FLAIR | |
|---|---|---|---|---|---|---|
| | PSNR↑ | SSIM↑ | PSNR↑ | SSIM↑ | PSNR↑ | SSIM↑ |
| Supervised PD-DL | 40.95 | 0.964 | 35.89 | 0.859 | 36.69 | 0.926 |
| SSDU | 40.12 | 0.956 | 35.35 | 0.856 | 36.98 | 0.929 |
| EI | 33.29 | 0.919 | 29.86 | 0.704 | 35.48 | 0.908 |
| ScoreMRI | 32.84 | 0.812 | 28.18 | 0.684 | 28.17 | 0.774 |
| CS | 36.71 | 0.917 | 32.30 | 0.749 | 32.93 | 0.865 |
| CG-SENSE | 35.38 | 0.873 | 30.13 | 0.702 | 30.25 | 0.801 |
| CUPID (**ours**) | 39.28 | 0.951 | 34.71 | 0.840 | 36.02 | 0.920 |

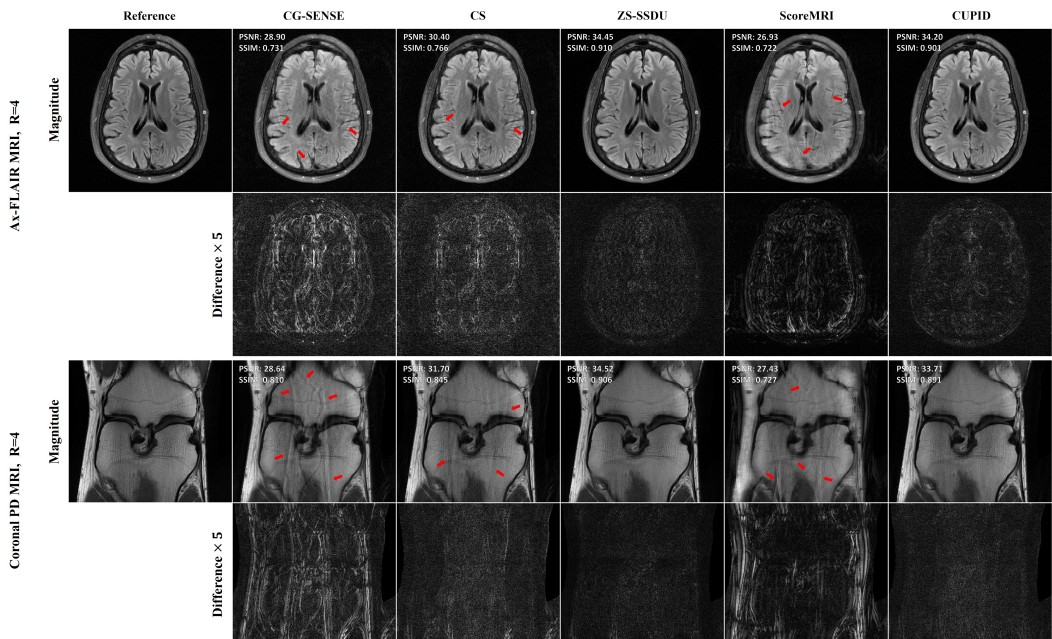

Figure 4: Representative subject-specific/zero-shot learning results for various algorithms on Ax-FLAIR and coronal-PD dataset for retrospective $R = 4$ uniform undersampling. Baseline CG-SENSE, conventional CS and ScoreMRI suffer from residual artifacts (red arrows). PD-DL with CUPID loss successfully removes these artifacts, and functions in a similar manner to ZS-SSDU.

these artifacts from the CG-SENSE image using a well-trained PD-DL network, achieving state-of-the-art reconstruction quality comparable to supervised PD-DL and SSDU, despite only having access to $x_{PI}$ for training, and not to raw k-space data unlike these other methods. We observe that parallel imaging reconstruction is not clinically usable at higher acceleration rates, but it is improved using a PD-DL reconstruction trained with CUPID. We further note that mild blurring was observed in some slices for database-training only. This is expected since we are no longer benefiting from redundancies from across multiple coils due to having no access to multi-coil raw k-space data, unlike the comparison methods. Quantitative results presented in Tab. 1 validate the visual observations, demonstrating that CUPID consistently outperforms CG-SENSE, CS, EI, and ScoreMRI across multiple datasets. Moreover, CUPID maintains performance comparable to that of supervised PD-DL and SSDU. Additional results on ax-FLAIR database are given in Appendix C.

**Zero-Shot Learning Results.** Fig. 4 shows results from zero-shot reconstructions. Both CG-SENSE and CS suffer from noise amplification and persistent residual artifacts, with CG-SENSE displaying a more pronounced degradation in quality. On the other hand, ScoreMRI exhibits blurring while still displaying residual artifacts. We note that uniform undersampling is used in these datasets, which is consistent with clinical parallel imaging acquisitions, but which have not been previously reported with diffusion models in existing works. Once again, CUPID demonstrates superior artifact and noise reduction over these methods and closely matches the quality of ZS-SSDU, despite not having access to raw data and an explicit self-validation mechanism to prevent overfitting as in the latter.

## 4.4 PRACTICAL SETTING: PROSPECTIVELY UNDERSAMPLED STUDY

As discussed in Sec. 4.1, brain data is acquired at the target acceleration rate, reconstructed via parallel imaging and exported in DICOM format to perform zero-shot fine-tuning. Fig. 5 shows reconstruction results for the vendor parallel imaging reconstruction, as well as CS, ScoreMRI and CUPID. CS reduces the noise in the parallel imaging reconstruction, but leads to blurring due to over-regularization. In contrast, ScoreMRI struggles to reconstruct accurately at this high acceleration rate, suggesting generalizability issues for the pre-trained score function to high-resolution imaging at a different field strength, not represented in the training database, and potential difficulties

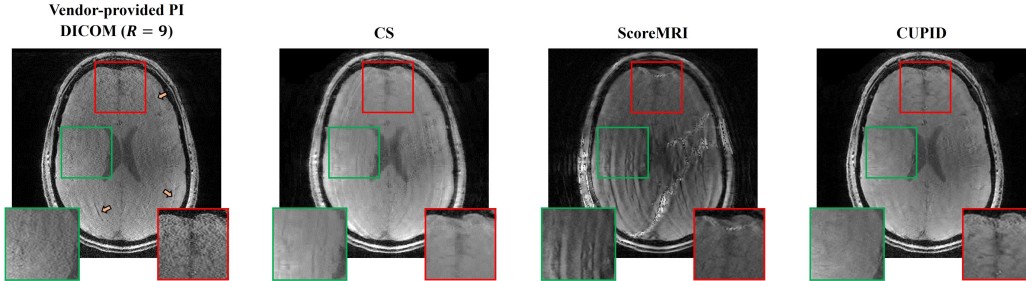

Figure 5: Prospective acceleration results for various methods that can operate on parallel imaging reconstructed DICOM images exported from the scanner. CUPID provides substantial noise and artifact reduction on the DICOM image, outperforming other methods.

with uniform undersampling. Furthermore, the public implementation of ScoreMRI uses hard constraints for data fidelity, which leads to more pronounced artifacts due to potential phase mismatch with the coils generated from separate calibration data. Our proposed CUPID method effectively mitigates both artifacts and noise in the DICOM image (shown in zoomed insets) without requiring any raw k-space data, attesting to the effectiveness of CUPID in real-world scenarios. Note minor residual artifacts remain since the target acceleration $R = 9$ in 1-dimension is very high. We note that ZS-SSDU cannot be applied here due to the unavailability of raw data. We further note that the vendor-provided DICOM was generated using k-space interpolation (Griswold et al., 2002) instead of the image domain formulation in (2). Due to their equivalence, this did not cause any issues for CUPID, as expected.

### 4.5 ABLATION STUDIES

We carried out two ablation studies to explore key factors influencing the performance of our algorithm. The first study explored the effect of $\lambda$ parameter to the final reconstruction, by training 5 distinct PD-DL networks using $\lambda \in \{0, 50, 100, 200, \infty\}$. We note that using $\lambda = 0$ corresponds to using only the compressibility term ($\mathcal{L}_{\text{comp}}$ in (8)), whereas using $\lambda \to \infty$ translates to using solely the parallel imaging fidelity term ($\mathcal{L}_{\text{pif}}$ in (9)). Fig. 6 shows the corresponding reconstruction results for each case. As outlined in Sec. 3, only using $\mathcal{L}_{\text{comp}}$ leads to overly-smooth reconstructions due to network forcing the wavelet coefficients towards zero without maintaining consistency with the data. On the other hand, solely using $\mathcal{L}_{\text{pif}}$ results in DIP-like reconstructions (Ulyanov et al., 2018), where the network overfits the data without any regularization, resulting in noise amplification. CUPID with $\lambda \in \{50, 100, 200\}$ integrates both loss terms to attain high-fidelity reconstructions. Thus, we conclude that CUPID demonstrates robust performance across a wide range of $\lambda$ values. Our second ablation study focuses on the effect of the number of perturbation patterns used in the training, and is provided in Appendix B.1.

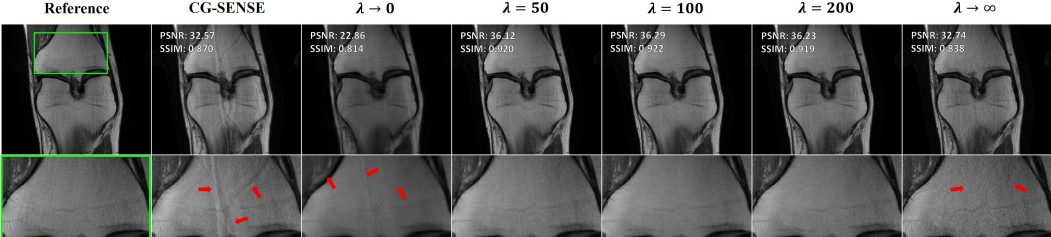

Figure 6: Using only the compressibility term ($\lambda = 0$) in the loss leads to overly-smoothed images, whereas using only the parallel imaging fidelity term ($\lambda \to \infty$) causes noise amplification (red arrows). Using both terms with a mid-range $\lambda$ value as a trade-off provides high-quality reconstructions that are clear from noise and artifacts.

4.6 DISCUSSION AND FUTURE WORK

**Filtering on Routine Clinical Images.** MR scans may include filtering operations applied by some vendors that affect the assumption $\hat{\mathbf{x}}_{\text{PI}} = (\mathbf{E}_\Omega^H \mathbf{E}_\Omega)^{-1} \mathbf{E}_\Omega^H \mathbf{y}_\Omega$. This was discussed extensively in Shimron et al. (2022), in the context of using retrospective undersampling of DICOM images to train DL reconstruction, especially highlighting the use of zero-padding, which improves the display resolution compared to the acquisition resolution. It was shown that training of models from retrospective undersampling of databases of DICOM images for PD-DL training using zero-padding may lead to biases and inaccuracies. Conversely, our approach is physics-driven in nature, and the sampling pattern $\Omega$ naturally accounts for the zero-padding operation. However, our method is not immune to other types of filtering/processing, such as implicit intensity correction (Han et al., 2001) or deidentification methods (Van Essen et al., 2013), in which case the filtered $\mathbf{x}_{\text{PI}}$ would need to be treated as the parallel imaging solution corresponding to a filtered version of $\mathbf{y}$.

**Resources for Fine-Tuning of PD-DL Reconstruction.** While our method is aimed to improve equitable access to fast MRI in low-resource settings, we do acknowledge that such low-resource MRI centers may lack the necessary hardware to fine-tune PD-DL models. We note that, similar to what has been shown in Yaman et al. (2022b), transfer learning along with zero-shot fine-tuning may be beneficial in decreasing training time and resources. Furthermore, since only DICOM images are needed for CUPID, standard anonymization techniques can be used (Van Essen et al., 2013), and data can be transferred to potential offsite locations with more computational resources. Since no raw data storage or transfer is necessary, the burden for transmission of DICOMs is low, and privacy can be preserved thorough anonymization, though specifics would need to be implemented with proper protocols.

**Magnitude-only DICOMs.** In some cases, MR vendors only give access to the magnitude of $\mathbf{x}_{\text{PI}}$ due to limitations in the reconstruction pipelines, for instance when using partial Fourier imaging in some vendors. Additionally, if a reference calibration scan is not prospectively acquired at exam time, coil sensitivities, and thus $\mathbf{E}_\Omega$ would be unknown as well, which is the case for most existing DICOM databases. In such cases, the phase of $\mathbf{x}_{\text{PI}}$ and coil sensitivities need to be estimated jointly. There has been work on the latter in the standard PD-DL setup with access to raw k-space data (Hu et al., 2024; Arvinte et al., 2021). However, this is a modification on the PD-DL objective function, and not the learning procedure considered here. Thus, it is not the focus of our study and it will be investigated in future studies.

**Reinforcement Learning for Optimal Perturbations.** Finally, a promising direction for future work may involve using reinforcement learning to simultaneously learn the optimal perturbations. This approach may enable the model to adapt and optimize perturbation strategies so that the sample mean estimate in (9) can be better approximated with a few perturbation examples.

## 5 CONCLUSION

In this study, we presented a novel training strategy, Compressibility-inspired Unsupervised Learning via Parallel Imaging Fidelity (CUPID), for PD-DL MRI reconstruction without access to raw k-space data. This approach leverages the compressibility of output images along with strategically designed perturbations that remain intact post-parallel imaging, thereby enhancing image quality in clinically accessible images in a physics-driven manner without the need for any raw k-space data. To the best of our knowledge, this is the first attempt that does not rely on raw data and uses these clinical images to train PD-DL networks, which is known for their high-fidelity reconstructions. CUPID also alleviates the training burden of generative methods, which requires a large number of data during training to capture the prior well. Quantitative and qualitative assessments of our method, conducted on both retrospectively and prospectively accelerated acquisitions, show its effectiveness in delivering high-quality performance across a diverse range of MRI scans and learning settings. Our source code will be released upon the acceptance of the paper.

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

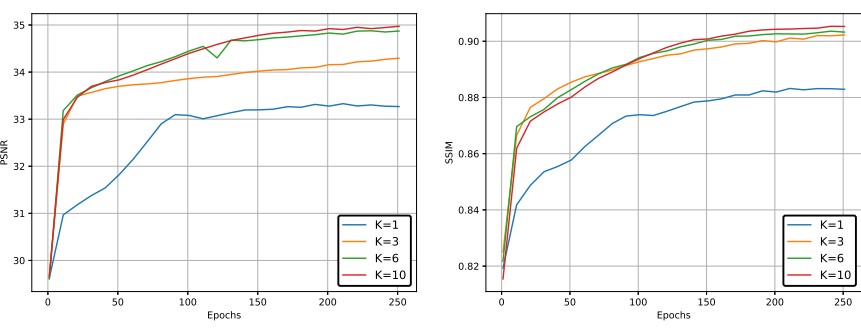

Figure 7: Representative zero-shot fine-tuning result for CUPID with different number of perturbations, $K \in \{1, 3, 6, 10\}$, on coronal PD knee MRI using $R = 4$ uniform undersampling. For $K < 6$, expectation in the second parallel imaging fidelity loss term can not converge to true mean, leading to artifacts in the final reconstruction. However, as $K$ increases, reconstruction quality also enhances. Note the gains from further increasing $K$ becomes negligible for $K > 6$.

## A  Implementation Details for Each Method

**Compressed Sensing.** We solved the regularized $\ell_1$ minimization problem given below:

$$\arg \min_{\mathbf{x}} \|\mathbf{y}_\Omega - \mathbf{E}_\Omega \mathbf{x}\|_2 + \tau \|\mathbf{W}\mathbf{x}\|_1, \tag{11}$$

using VS-QP (Fessler, 2020). Similar to the unrolled network, data fidelity was solved using CG, and soft thresholding was implemented on the DTCWT coefficients.

**ScoreMRI.** For ScoreMRI implementation, we followed the original code and pre-trained network provided by Chung & Ye (2022) in their corresponding public repository.

**PD-DL Based Approaches.** For each method, the unrolled network comprised 10 unrolls, while the regularizer was implemented as a CNN-based ResNet architecture (Timofte et al., 2017) that had 10 residual blocks. Data fidelity was achieved using a conjugate-gradient (CG) method (Aggarwal et al., 2019) with 10 iterations. The unrolled network was trained in an end-to-end fashion for 100 epochs. For supervised PD-DL (Hammernik et al., 2018; Aggarwal et al., 2019), the normalized $\ell_1$-$\ell_2$ loss function was used between the reconstructed and ground truth raw k-space data (Knoll et al., 2020a). For SSDU, $\rho = |\Delta|/|\Omega| = 0.4$ was used as proposed in Yaman et al. (2020). For EI (Chen et al., 2021), we modified the loss function in PD-DL networks to:

$$\min_{\boldsymbol{\theta}} \mathbb{E}\left[\mathcal{L}\left(\mathbf{y}_\Omega, f(\mathbf{y}_\Omega, \mathbf{E}_\Omega; \boldsymbol{\theta})\right)\right] + \beta \sum_{g \in G} \mathcal{L}\left(\mathcal{T}_g f\left(\mathbf{y}_\Omega, \mathbf{E}_\Omega; \boldsymbol{\theta}\right), f\left(\mathbf{E}_\Omega \mathcal{T}_g f\left(\mathbf{y}_\Omega, \mathbf{E}_\Omega; \boldsymbol{\theta}\right), \mathbf{E}_\Omega; \boldsymbol{\theta}\right)\right) \tag{12}$$

in which the first term enforces consistency while the second term imposes equivariance relative to a group of transformations, $\{\mathcal{T}_g\}_{g \in G}$. Here, $|G|$ defined as the cardinality of $\{\mathcal{T}_g\}_{g \in G}$ and $\beta$ is the equivariance weight. We followed the authors' publicly available CT reconstruction code

(a) PSNR curves for each $K$ value.     (b) SSIM curves for each $K$ value.

Figure 8: PSNR and SSIM curves confirm the visual observations with respect to the number of perturbations, $K$. Lower $K$ values tend to perform worse and increasing $K$ becomes redundant after a certain point.

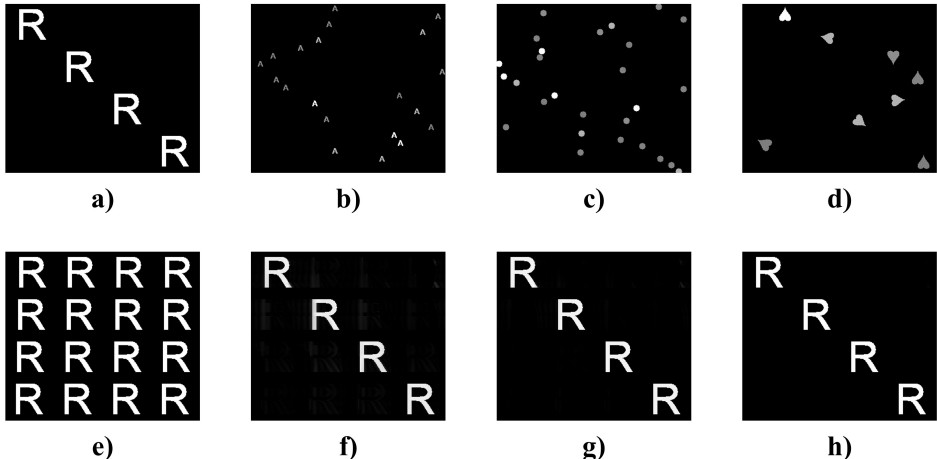

Figure 9: Added perturbations may consist of: (a) precisely positioned letters that have the same intensity in within each shape, (b) randomly positioned letters or (c) circles that have different intensities, or (d) randomly rotated card suits. Furthermore, when accelerated by $R = 4$, even without ACS data (e), these perturbations can be resolved via parallel imaging methods such as CG-SENSE: (f) 20 iterations, (g) 40 iterations, (h) 80 iterations.

for EI (Chen et al., 2021), and employed 3 rotations along with 2 flips. For CUPID, dual-tree complex wavelet transform (DTCWT) which provides an over-complete representation (Selesnick et al., 2005; Cotter, 2020) was selected as the sparsifying transform ($\mathbf{W}$) in (8). Furthermore, $\mathbf{x}^{(0)}$ in (8), i.e. the initial estimate prior to any reweighting, was calculated using a CS approach as mentioned in Section 3 This was implemented using (11) with 100 iterations, with 30 CG steps for data fidelity and $0.1 \cdot ||\mathbf{W}\mathbf{x}_{\text{PI}}||_\infty$ as the soft thresholding parameter.

## B  PERTURBATION STRATEGIES

### B.1  CHOICE FOR NUMBER OF PERTURBATION PATTERNS

The empirical expectation that approximates the one in (9) is expected to converge to the true expectation as we introduce more perturbation patterns and randomness over the choice of $\mathbf{p}$. Fig. 7 shows the zero-shot fine-tuning results of CUPID with $K \in \{1, 3, 6, 10\}$, while Fig. 8(a) and Fig. 8(b) illustrates the corresponding PSNR and SSIM curves throughout the training epochs, respectively. As expected, using a single pattern could not capture the true mean and exhibits artifacts. As we introduce more perturbations, we reduce the artifacts and noise amplification. At a certain point, increasing the number of perturbations becomes counterproductive, yielding only marginal gains while significantly increasing the computation time. Thus, we opted to use 6 distinct $\mathbf{p}_k$ patterns throughout our study as it offers the optimal trade-off.

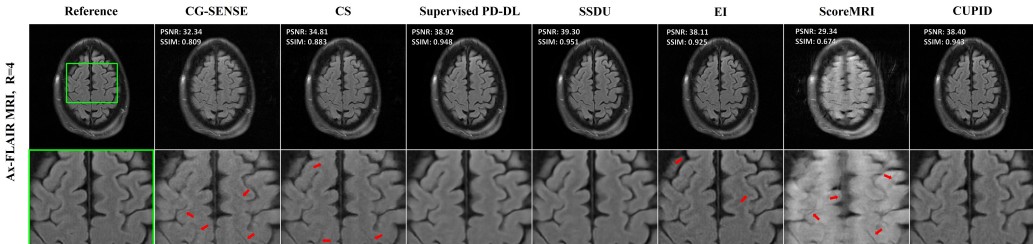

Figure 10: Representative ax-FLAIR database reconstructions based on various learning and reconstruction strategies. CG-SENSE, CS, EI and ScoreMRI exhibit artifacts. On the other hand, CUPID surpasses them with high-fidelity reconstructions, closely matching supervised and self-supervised methods that requires raw k-space data during training.

Table 2: Quantitative results with standard error of the mean included. Each method is tested on Coronal PD, Coronal PD-FS and Ax-FLAIR datasets using equispaced undersampling pattern at $R = 4$.

| Method | Coronal PD | | Coronal PD-FS | | Ax-FLAIR | |
|---|---|---|---|---|---|---|
| | PSNR↑ | SSIM↑ | PSNR↑ | SSIM↑ | PSNR↑ | SSIM↑ |
| Supervised | $40.95 \pm 2.95$ | $0.964 \pm 0.016$ | $35.89 \pm 2.72$ | $0.859 \pm 0.089$ | $36.69 \pm 1.73$ | $0.926 \pm 0.016$ |
| SSDU | $40.12 \pm 2.95$ | $0.956 \pm 0.019$ | $35.35 \pm 2.73$ | $0.856 \pm 0.091$ | $36.98 \pm 1.83$ | $0.929 \pm 0.016$ |
| EI | $33.29 \pm 6.65$ | $0.919 \pm 0.034$ | $29.86 \pm 6.59$ | $0.704 \pm 0.156$ | $35.48 \pm 4.55$ | $0.908 \pm 0.028$ |
| ScoreMRI | $32.84 \pm 4.48$ | $0.812 \pm 0.035$ | $28.18 \pm 4.25$ | $0.684 \pm 0.142$ | $28.17 \pm 3.84$ | $0.774 \pm 0.034$ |
| CS | $36.71 \pm 2.86$ | $0.917 \pm 0.036$ | $32.30 \pm 2.68$ | $0.749 \pm 0.158$ | $32.93 \pm 2.03$ | $0.865 \pm 0.028$ |
| CG-SENSE | $35.38 \pm 3.04$ | $0.873 \pm 0.039$ | $30.13 \pm 2.73$ | $0.702 \pm 0.164$ | $30.25 \pm 2.12$ | $0.801 \pm 0.029$ |
| CUPID (**ours**) | $39.28 \pm 2.91$ | $0.951 \pm 0.022$ | $34.71 \pm 2.61$ | $0.840 \pm 0.093$ | $36.02 \pm 1.70$ | $0.920 \pm 0.018$ |

### B.2 DESIGN ALTERNATIVES FOR PERTURBATIONS

As stated in Sec. 3, added perturbations may consist of several different structures. Fig. 9 provides some of these perturbation examples, an illustration of how the perturbation looks with undersampling, and how they are recovered perfectly through conventional parallel imaging methods. We note that there was no task-specific perturbation that we used, meaning that the perturbations selected from the same set were applied to all datasets given that the created perturbations do not create fold-overs at R-fold which result in artifacts. Note the latter condition means they should be recoverable through parallel imaging reconstruction. Finally, we note that when calculating the sample mean estimate for (9), intensity of the perturbations was empirically found to be more important than their shapes/orientations. Specifically, we observed that varying it randomly within the perturbation, as in Fig. 9b-d, leads to improved reconstruction outcomes.

## C MORE RESULTS FROM THE RETROSPECTIVE STUDY

We further include the representative reconstructions from the Ax-FLAIR dataset in Fig. 10.

## D FURTHER QUANTITATIVE RESULTS

A more detailed version of Tab. 1, incorporating the standard error of the mean, is provided in Tab. 2.

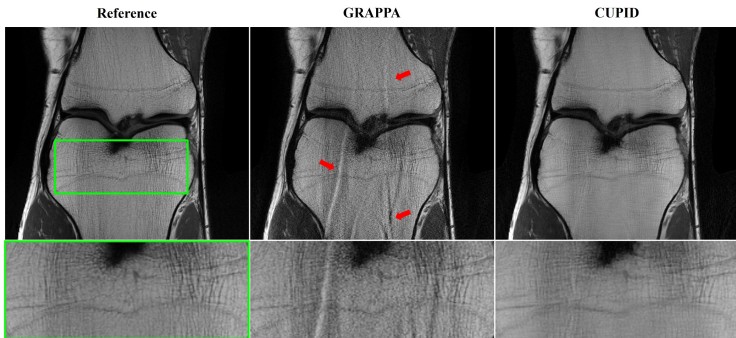

Figure 11: Representative reconstructions for CUPID with $\mathbf{x}_{PI}$ reconstructed using GRAPPA on coronal PD knee MRI using $R = 4$ uniform undersampling. GRAPPA exhibits aliasing and noise artifacts at this high acceleration rate. PD-DL network trained with a CUPID implementation that only has access to this GRAPPA reconstruction improves on it, reducing these artifacts. This highlights the compatibility of CUPID with different parallel imaging reconstructions.

# E    COMPATIBILITY WITH VARIOUS PARALLEL IMAGING RECONSTRUCTIONS

Vendor reconstructions typically use different parallel imaging techniques. For our retrospective studies, we used CG-SENSE (or equivalently SENSE) (Pruessmann et al., 1999) because it naturally fits with the DF units in the unrolled network, and it is commonly used in clinical settings, alongside GRAPPA (Griswold et al., 2002). However, we emphasize that our method does not make assumptions about the specific reconstruction method used by the vendor; instead, it assumes that parallel imaging can resolve the perturbations, which is ensured by designing them in a manner that prevents fold-over aliasing artifacts from overlapping.

To further validate this, we include representative CUPID reconstruction results in Fig. 11 where $x_{PI}$ is generated via GRAPPA (Griswold et al., 2002), demonstrating that CUPID is compatible with different types of parallel imaging reconstructions as input. We further note that the prospective study also used GRAPPA reconstruction as input, as this is the reconstruction provided by the vendor used in our institution.

