# OpenReview forum: "Training Physics-Driven Deep Learning Reconstruction without Raw Data Access for Equitable Fast MRI"
_ICLR.cc/2025/Conference — Submitted to ICLR 2025_

### Official Review · Reviewer_snDg · 2024-10-28

**Soundness:** 3
**Presentation:** 3
**Contribution:** 2
**Rating:** 5
**Confidence:** 4

**Summary:**

This paper proposes a loss formulation for unsupervised MRI reconstruction that directly operates on under-sampled DICOM images, bypassing the need for raw k-space measurements. The proposed loss involves a reweighted L1 loss term that enforces compressibility of the output image and a consistency loss term under designed perturbations to prevent trivial solutions. The method is shown to also work in a scan-specific setting. The experiments were conducted on knee and bran scans of fastMRI datasets with 4x acceleration and equidistant undersampling patterns, as well as prospective undersampled images.

**Strengths:**

- The loss design is reasonable.
- The overall method is succinct and achieves very competitive performance.
- Evaluated the method on both retrospectively and prospectively undersampled datasets.

**Weaknesses:**

1: In [1], equivariance to image perturbations is also enforced alongside the measurement consistency term. Given the similarity, what makes the proposed method work so much better than [1] ?
2: The authors mention that the overfitting issue here is not as severe as in other self-supervised scan-specific methods due to $L_{comp}$. However, $L_{comp}$ is also associated with some important parameters that are subject to manual tuning. It's not clear how those choices would affect overfitting.

[1] "Equivariant Imaging: learning beyond the range space." Chen et al., ICCV 2021.

**Questions:**

See above.

---

> ### Author Response · Authors · 2024-11-21
> **Response to Reviewer snDg**
>
> **W1. Improvement over equivariant imaging (EI)**
>
> **A.** We thank the reviewer for this valuable comment. In [1], the authors consider transformations, such as shifts, rotations, or reflections rather than added perturbations which is a different approach to CUPID. We further note that our approach also differs from [1] in terms of access to raw k-space data, as [1] assumes such access is available, whereas our method does not. For EI implementation, we applied three rotations, similar to the approach used by the authors in their publicly released code for the CT application, and additionally incorporated two flips. Notably, the authors did not use shifts in their CT application, and we found that shifts were not useful in the MRI context either, as they only modify the phase in k-space. We now detail these aspects further in Appendix A.
>
> Another difference between the two approaches is that [1] does not enforce sparsity alongside this transformation loss, which can further increase the performance gap between EI and CUPID. To further highlight this point, in Figure 6, we showed that only using CUPID's fidelity operator ($\mathcal{L}\_{pif}$) without $\mathcal{L}_{comp}$ (which accounts to $\lambda \rightarrow \infty)$ also results in subpar results with noise amplification and minor artifacts. Thus, the combination of both our terms is needed for improved performance.
>
> **W2. Overfitting in CUPID**
>
> **A.** We thank the reviewer for this valuable comment. The only hand-tuned parameters for $\mathcal{L}\_{comp}$ are $\mathbf{x}^{(0)}$ (the initial input prior to reweighting) and $\epsilon$. Here, $\epsilon$ is a very small number added for numerical stability. Specifically, this $\epsilon$ is added to prevent excessively large losses that may result from a zero, or values close to zero, in the denominator. On the other hand, we acknowledge that there are some parameters in creation of $\mathbf{x}^{(0)}$ that is hand-tuned, such as soft-threshold limit or data fidelity iteration number. Nevertheless, this still does not change the fact that $\mathcal{L}_{comp}$ does not involve a subtraction between two entities. Thus, if one were to substantially change these parameters in generating $\mathbf{x}^{(0)}$, they would not lead to overfitting, but rather to poor reconstructions.
>
> Additionally, the objective in creating $\mathbf{x}^{(0)}$ is to produce the sparsest image possible that is free of artifacts, though it appears highly blurry. We have added the hyperparameters associated with the generation of $\mathbf{x}^{(0)}$ in Appendix A. We thank the reviewer for pointing out this oversight.

---

> > ### Author Response · Authors · 2024-11-23
> > **A Gentle Reminder**
> >
> > Dear Reviewer snDg,
> >
> > We sincerely thank you for taking the time to review our responses and for your valuable feedback throughout this process. As the rebuttal period is nearing its conclusion, we hope that our revisions and clarifications have adequately addressed the issues you highlighted. If you find our responses satisfactory, we would be truly grateful if you could consider revisiting your initial evaluation.
> >
> > Should there be any remaining points or areas requiring further clarification, please do not hesitate to let us know. We are more than willing to make additional adjustments as needed.
> >
> > Thank you once again for your time and effort.
> >
> > Kind regards,
> >
> > Authors

---

### Official Review · Reviewer_CGvy · 2024-11-01

**Soundness:** 2
**Presentation:** 1
**Contribution:** 2
**Rating:** 3
**Confidence:** 5

**Summary:**

This paper present an novel MRI reconstruction method that only requires dicom data (even undersampled data), with the motivation that rural regions won't have the raw data access. The key idea is through using random perturbations.

**Strengths:**

1. I like the idea of using random image domain perturbations, sounds like a good way to capture pior distribution.
2. I appreciate the authors for the experiments on the prospective undersampled MRI reconstruction, which is a very important experiment for all MRI reconstruction tasks.

**Weaknesses:**

Overall, there are a few weaknesses that concerns me:
1. I didn't really buy the idea of this rural region raw-data access. why we need raw-data? its for the training purpose! we already have a great number of raw k-space data (from fastMRI, mridata.org and others), as long as we have a good model trained, the rural regions can just use it without any training. In the end of the day, all MRI scanner will sample raw k-space data, so the whole motivation doesn't sound solid to me.
Meanwhile, from what i understand from your method, the input is CG-SENSE of undersampled MRI, is it realistic? the input quality is not really diagnoistic.

2. The presentation is not very clear and the paper is not well-writtened, all the method section for your method is condensed to one *single* page with a figure (figure 2), i have to go back and forth to understand whats R, whats DF, and how you do the perturbations, thats supposed to be the main part of the paper.

3. The score-MRI results look kinda weird to me, try to look at other generative models. It shouldn't be that bad.

4. Figure 3, the result doesn't look good to me, over smoothing textures.

**Questions:**

1. how do your method compared to single-coil DL-based reconstruction.
2. how sensitive is your method to different input? (different vendors tend to have different reconstruction pipelines.)

---

> ### Author Response · Authors · 2024-11-21
> **Response to Reviewer CGvy (Part 1)**
>
> **W1. "I didn't really buy the idea of this rural region ..."**
>
> **A.** We thank the reviewer for bringing up this point, as it looks like our initial exposition was unclear. The use of DL-based reconstruction in clinical environments have been limited due to issues related to generalizability and artifacts. Various studies have noted this generalizability problem to be caused by mismatches in training data and the target population data. Specifically, the model performs poorly when exposed to test samples that fall outside the distribution of the training set. For instance, consider the following statements from a recent publication in the high-profile journal Science (DOI: 10.1126/science.adm7168) on portable MRI techniques that use DL-based methods:
>
> *"However, the fidelity of the PF-SR method in restoring 3D image details remains to be carefully evaluated and optimized for each anatomical structure and contrast. [...] In fact, hallucinations can be seen among some sulci and gyri near the brain edge in the T1W PF-SR results [...] This prior knowledge is deeply ingrained within the PF-SR models, which are trained to learn the structural and contrast 3D multiscale features from a large collection of standard human MR images specific to a particular organ and MRI contrast. Future research should also optimize and evaluate the capabilities of PF-SR in detecting various pathologies."*
>
> A current popular solution to overcome this issue is to fine-tune/retrain the model in the target population, and such works have been published in ICLR before. This ties into the problem we are trying to solve: For numerous target populations, the raw data is not available at all, since the MRI centers do not have research agreements in place. This is even more pronounced in rural/underserved environments, as virtually no MRI centers in these settings have raw data access. Therefore, even though we have some great pre-trained models that use raw k-space data, the absence of a method that can train or further fine-tune the model over a few or a single subject without accessing the raw k-space data is a shortcoming that needs to be addressed, as also noted by the other reviewers.
>
> **W1. "In the end of the day, all MRI scanner will sample raw k-space data, ..."**
>
> **A.** We kindly disagree with the reviewer on this subject. While it is true all MRI scanners will sample raw k-space data, as noted early, unfortunately, majority of MRI scanners outside specialized/academic MRI centers, *e.g.*, those in local hospitals or mobile MRI units, do not offer raw k-space access/export capabilities because of the absence of vendor agreements. Thus, any fine-tuning method needs to be compatible with DICOM images.
>
> **W1. "from what i understand from your method, the input is CG-SENSE of undersampled MRI, is it realistic? the input quality is not really diagnoistic"**
>
> **A.** It is true that the parallel imaging reconstruction will be clinically unusable at the higher acceleration rates, and this is the problem our method is trying to address. As we show in the manuscript, the clinical DICOM images that have artifacts due to parallel imaging reconstruction can be improved using a PD-DL approach trained using CUPID.
>
> **Q2. "how sensitive is your method to different input? (different vendors tend to have different reconstruction pipelines.)"**
>
> **A.** All vendor reconstructions use some type of parallel imaging method. For our retrospective studies, we opted to use CG-SENSE (or equivalently SENSE) as it is the most commonly used clinical method along with GRAPPA. But our method does not assume anything about how the reconstruction is generated, it only assumes that the perturbations can be resolved by a parallel imaging method, which is ensured by not allowing fold-over artifacts to overlap *by design*. We modified Fig. 9 by adding another row (e-h) to further clarify this point.
>
> Furthermore our prospective study is from vendor data that uses GRAPPA, and our method works well in this setup as well. This data also had the aforementioned enhancements (regularization, filtering), proving CUPID's compatibility with different input types. We have now included further results as part of a retrospective study in Appendix E, where ${\bf x}_{PI}$ is derived from GRAPPA. We appreciate the reviewer for raising this point, as we believe it has strengthened our manuscript.

---

> > ### Author Response · Authors · 2024-11-21
> > **Response to Reviewer CGvy (Part 2)**
> >
> > **W2. ", all the method section for your method is condensed to one single page with a figure (figure 2), i have to go back and forth to understand whats R, whats DF, and how you do the perturbations, thats supposed to be the main part of the paper"**
> >
> > **A.** We kindly note we detailed how the perturbations are done in the ``Parallel Imaging Fidelity'' subsection on p. 5, around Eq. 9. We have highlighted this part for easier reference. We have added further details in Appendix B and Figure 9 that goes into specifics.
> >
> > We apologize for not defining the abbreviations, R (regularizer) and DF (data fidelity) in their first use. These are now defined in the figure caption. All other abbreviations are also defined properly.
> >
> > **W3. Score MRI results**
> >
> > **A.** We thank the reviewer for this valuable comment. We acknowledge there is a performance difference compared to the original publication. For our experiments, we used the pre-trained score model that is provided by the authors in their public repository*. However, we have since noticed that they deleted their pre-trained network after we downloaded it and there is no way to reach the file right now. Thus, we suspect there may be differences between the model used in the ScoreMRI paper and the pre-trained model that was released.
> >
> > That being said, we do not think it would be appropriate for us to train this model from scratch and implement it, as it would not be fair to the ScoreMRI authors. Based on Rewiever 4Fqf's comments, we are currently looking into some other generative models and if we can not fix the problem with Score MRI, we will remove its results from our manuscript.
> >
> > We would be happy to include a newer approach here - however we are not aware of any recent works that have publicly available pre-trained score models for MR images. If the reviewer can recommend such a work, we will do our best to update the comparisons by the end of the discussion period.
> >
> > *https://github.com/HJ-harry/score-MRI/tree/main
> >
> > **W4. "Figure 3, the result doesn't look good to me, over smoothing textures."**
> >
> > **A.** We kindly note that even though our method demonstrates improvements over approaches that do not utilize raw k-space data, it is expected to lag behind methods that do in some aspects, as it operates solely on DICOM images acquired *at the target acceleration rates*. We observed that even though it smooths a few slices on database training, it delivers sharp reconstructions in zero-shot/subject-specific settings (Fig. 4 and Fig. 5). However, it is important to note that in both database-training and zero-shot training cases, CUPID successfully removes all aliasing artifacts, present in the baseline parallel imaging (i.e. DICOM) images. Finally, we also highlight that while we are achieving competitive results to SOTA, we are no longer benefiting from redundancies from across multiple coils due to having no access to raw k-space data. We modified our manuscript to mention the blurring artifacts in the database case, and to emphasize the data availability difference to SOTA methods.
> >
> > **Q1. Single-coil DL-based reconstructions**
> >
> > **A.** Most of the PD-DL reconstruction focuses on regularizing a coil-combined image acquired with multi-coil datases. We do acknowledge there are a few works that perform coil-by-coil reconstruction, such as Score MRI, but this is not the norm. Furthermore, many standard clinical scans do not allow exporting DICOM data coil-by-coil$^*$, thus it is not applicable to the scenario we are interested.
> >
> > $^*$: For coil estimation in the prospective case, we did use such a scan for scout images (as described on p. 7). But this option to export coil-by-coil DICOMs is not available for many high-resolution clinical scans, at least on the vendor we use.

---

> > > ### Author Response · Authors · 2024-11-23
> > > **A Gentle Reminder**
> > >
> > > Dear Reviewer CGvy,
> > >
> > > We sincerely thank you for taking the time to review our responses and for your valuable feedback throughout this process. As the rebuttal period is nearing its conclusion, we hope that our revisions and clarifications have adequately addressed the issues you highlighted. If you find our responses satisfactory, we would be truly grateful if you could consider revisiting your initial evaluation.
> > >
> > > Should there be any remaining points or areas requiring further clarification, please do not hesitate to let us know. We are more than willing to make additional adjustments as needed.
> > >
> > > Thank you once again for your time and effort.
> > >
> > > Kind regards,
> > >
> > > Authors

---

> > > ### Comment · Reviewer_CGvy · 2024-11-25
> > > **Finetuning experiment**
> > >
> > > I appreciate the authors for their feedback, which addressed some of my concerns.
> > >
> > > regarding the W1 and A, you mentioned fine-tuning experiment for different scanners, why this is crutial? do you have any experimnt to support the claim - finetuning on different scanners is nessecary.

---

> ### Author Response · Authors · 2024-11-26
> **Official Comment by Authors (Part 1/2)**
>
> We thank the reviewer for raising this point. Although pre-trained networks that leverage k-space data during training are available, they often face challenges with generalizability or hallucinations. Specifically, such networks struggle to reconstruct MRI data that was not included in their training set, particularly when there are differences in SNR, acquisition protocols, or the scanned organ. We also note the following distinction, which seems to have caused confusion: In the text, we do not talk about fine-tuning on different scanners, but talk about fine-tuning to handle out of distribution examples. The issues related to out of distribution generalization are well-documented in the literature. For instance:
>
> - In doi: 10.1002/mrm.27355: *"One of the biggest open questions regarding the success
> of these technologies in practice is generalization. To what degree can the test data deviate from the data that were used during training? This is important for several reasons. First, one of the key strengths of MRI is the flexibility during data acquisition. Due to the range of available MR systems and protocols, images from different institutions commonly vary with respect to acquisition parameters. [...] a deviation of SNR between training and test data leads to a substantial reduction of image quality when using a trained variational network for image reconstruction."*
>
> - Specifically for scanner differences, in doi: 10.1109/TMI.2021.3075856: *For the Transfer track, participants were asked to run their models on data from vendors outside the main fastMRI data set. There was a caveat: we also restricted participants in the Transfer track to train their models only using available fastMRI data to ensure evaluation of transfer capability. [...] We observed decreases in performance in the Transfer track. Many participants struggled to adapt their models to [vendor] data [...] We note that as designed the Transfer track primarily evaluated one type of transfer: generalization across vendors.*
>
> - In doi: 10.1007/978-3-030-88552-6_3, *All of these methods achieved high image quality but their robustness to possible domain shifts between training and test data remained an open question. In real world clinical use, the images may have unique features not seen in the training data, and they may vary in terms of SNR and coil configuration. Our results show that all of the methods remove small structures not seen in the training data and generate over smoothed images when the model input has lower SNR.*
>
> Here, the unique features not seen on training data is important, as it is a surrogate for pathologies that are not well-represented in the training database, but may be present in the target population. This is where the difference between the population demographics between MRI centers that have raw data access (typically urban/academic) and that do not have raw data access (local community hospitals, mobile MRI units) become important, which is what our paper tackles.
>
> - And in a review article from this year doi:10.1007/s10334-024-01173-8, *Good generalization of DL-based MRI reconstruction models is critical for clinical workflows. However, achieving a good generalization is challenging because MRI data can vary substantially in terms of different factors, e.g. MRI hardware, vendor-specific scanning protocols, patient populations, and the anatomical regions being imaged. This variability can lead to a model that performs well on data from one source but poorly on data from another, a phenomenon known as domain shift or distribution shift. [...] Domain shifts have been studied from several perspectives. Johnson et al. analyzed the robustness of the models submitted to the 2019 fastMRI challenge to distribution shifts, e.g. small structural changes, addition of noise to k-space data, and changes in the number of coils. The study found that many of these models were sensitive to the distribution shifts. Darestani et al. evaluated the robustness of DL reconstruction methods with regard to out-of-distribution data, and found that both trained and untrained networks were affected by distribution shifts. Avidan et al. studied another type of distribution shift, related to sampling; methods trained on specific sampling schemes may not generalize well to other schemes. Altogether, distribution shifts can lead to substantial performance drops in MRI and can hence be a major limiting factor in practice.*

---

> ### Author Response · Authors · 2024-11-26
> **Official Comment by Authors (Part 2/2)**
>
> and also the following:
>
> *However, achieving a good generalization is challenging because MRI data can vary substantially in terms of different factors, e.g. MRI hardware, vendor-specific scanning protocols, patient populations, and the anatomical regions being imaged. This variability can lead to a model that performs well on data from one source but poorly on data from another, a phenomenon known as domain shift or distribution shift. [...] The term hallucinations refers to the generation of false, realistic-looking features which are not present in the actual data. This can arise from the use of inaccurate priors, *e.g.*, when there is a distribution shift between the training and test data, as described above. Strikingly, the team that organized the second FastMRI challenge found that many of the top-performing models produced hallucinations, and that these hallucinations were not captured by image quality metrics such as SSIM. They also noticed that hallucinations could morph abnormal structures into seemingly normal ones.*
>
> In fact, several illustrative examples of such shifts have been published in papers that discuss test-time training, such as Darestani et al, ICML, 2022, and Yaman et al, ICLR, 2022.
>
> Importantly, we note the following statements from the last article on mitigation for these issues, which aligns with what we describe in our paper: ***Potential mitigation strategies.** In cases where only a few training examples from a target domain are available, pre-training a network on other data and fine-tuning it to the target domain can improve performance. In the challenging case where no target data are available for fine-tuning, test-time-training, which involves adapting to a single training at inference, is a viable performance-enhancing alternative. Another good strategy that can help mitigate the performance drop due to distribution shifts is to train on broad and diverse data.*
>
> **These are the challenges that CUPID seeks to address**. Our method allows this fine-tuning to be done on DICOM images acquired at target acceleration rates without raw k-space access. Thus, it can be used at MRI centers that do not have agreements for raw k-space access, paving the way for improving the access to such DL-based reconstruction.
>
> We once again thank the reviewer for asking us to further explain these concepts. Due to space issues, our description in the text was inherently succinct. But with the discussion here, we were able to clarify related challenges in MRI reconstruction in a more accessible manner.

---

### Official Review · Reviewer_4Fqf · 2024-11-03

**Soundness:** 3
**Presentation:** 3
**Contribution:** 3
**Rating:** 6
**Confidence:** 4

**Summary:**

The article proposes a deep-learning reconstruction method without having access to multi-coil $k$-space data. For the full-column-rank forward operator $E_{\Omega}$ and the under-sampled $k$-space $\textbf{y}$, the authors assume access to $E_{\Omega}^+ \textbf{y}$ where $+$ denotes the Moore-Penrose inverse. Compared to having access to fully sampled $k$-space as in fully supervised reconstruction, or undersampled $k$-space as in self-supervised reconstruction, this assumption is less restrictive. The proposed method was shown to perform on par with PD-DL methods requiring $k$-space data at relatively low acceleration rates ($4\times$).

**Strengths:**

1. The data access assumption is less restrictive compared to fully- and self-supervised reconstruction methods, and the article focuses on a real-life problem, as raw k-space access is a known limitation in this field.
2. The method is physics-based; thus, it is expected to be less prone to hallucinations compared to methods that do not utilize the forward operator.
3. The method was tested in various settings, focusing on both retrospective and prospective undersampling.

**Weaknesses:**

1. The quantitative results table is missing the standard error of the mean, which is crucial for a fair comparison.
2. It is incorrect to say that equispaced undersampling patterns were not explored in score-based models. In fact, the authors have already cited at least one article ("Robust Compressed Sensing MRI with Deep Generative Priors") that explored equispaced undersampling, discussing it not only in the appendix but also in the main text (e.g., Fig. 2, 9, 10).
3. The generative method baseline is weak and outdated, given how rapidly state-of-the-art methods are advancing.  Several approaches published in the last 12 months can achieve better results. Including this particular method is fine, but labeling it as state-of-the-art is misleading.

**Questions:**

1. Does the clinical PI reconstruction readily allow saving the complex-valued image without any research agreements, or does it only allow the magnitude image to be saved? I am asking because it is rare to see a DICOM dataset containing anything other than magnitude images.
2. Regardless of whether ScoreMRI is state-of-the-art, its performance appears unexpectedly low. Is it possible that it was significantly under-trained? In our experience, these models require much longer training times compared to unrolled networks, which might explain why it underperforms even at $4\times$.
3. The choice of perturbations seems somewhat arbitrary. Could you elaborate on this choice? Learning the perturbations simultaneously with reinforcement learning could also be a potential direction for future work.
4. In Figure $2$, loss function, the left hand side was written as $\mathcal{L}(\mathbf{x}^{(k)}, \mathbf{x_{PI}})$ but the $\mathcal{L}_{\text{comp}}$ on the right hand side has $\mathbf{x}^{(m)}$ rather than $\mathbf{x}^{(k)}$. Is this intentional or a typo?
5. Line $151-152$, while unrolled methods achieved top positions in reconstruction challenges three years ago, it is unknown if they remain the highest performers. To the best of my knowledge, no recent, fair comparison between state-of-the-art diffusion and unrolled methods has been published, including a reader study to compare their clinical effectiveness.

**Minor comments**

* The fastMRI knee matrix size is $320\times 320$, not $320\times 368$. Please see Table 1 in DOI: 10.1148/ryai.2020190007
* Line $42-43$, demand have shown -> demand has shown
* Line $269$, shapes that has different intensity values -> shapes that have different intensity values
* Line $46$, Fig.$1$ -> Fig. $1$, the space was omitted in most, if not all, references to figures, tables, and sections.

---

> ### Author Response · Authors · 2024-11-21
> **Response to Reviewer 4Fqf**
>
> **W1. Standard error of the mean in the table**
>
> **A.** We thank the reviewer for their efforts in improving our work. The full table including the standard error of the mean is now added in the Appendix D. We have kept the simpler version in the main text for easier readability.
>
> **W2. Exploration of equispaced undersampling in terms of generative models**
>
> **A.** We thank the reviewer for pointing this out, we removed this statement from our manuscript.
>
> **W3 \&  Q2. Regarding Score MRI**
>
> **A.** *We completely agree with the reviewer, and* we would be happy to include a newer approach here - however, we are not aware of any recent works that have publicly available pre-trained score models for MR images. If the reviewer can recommend such a work, we will do our best to update the comparisons by the end of the discussion period.
>
> We note that we do not train ScoreMRI from scratch here, as we think it would not be fair to the original authors given that it requires a large database and resources to train. Therefore, we suspect that the pre-trained score model that is provided by the authors in their public repository* may be the problem, as we noticed that they deleted their pre-trained network after we downloaded it and there is no way to reach the file right now.
>
> *https://github.com/HJ-harry/score-MRI/tree/main
>
> **Q1. Saving the complex valued DICOMs**
>
> **A.** We thank the reviewer for raising this point. It depends on the vendor and the sequence. For most cases, they do, but there are cases where it is not available, e.g. when using partial Fourier imaging on certain vendors. We discuss these points and other issues for extending to magnitude-only DICOMs on p. 9.
>
> **Q3. Choice for perturbations**
>
> **A.** The main focus while crafting these perturbations was to ensure that the R-fold aliasing does not overlap within the field of view, allowing it to be resolved using parallel imaging reconstruction. Specifically, no other shape appears in the phase-encoding direction, after undersampling, in Figure 9. We have now updated Figure 9 with an additional row (e-h) to highlight how this works with aliasing and how the aliasing is resolved by parallel imaging methods. We hope this clarifies the process further.
>
> The other aspect is the expectation in Eq. 9. As mentioned in the text, we implemented this by a sample mean, as is usually done, and used random orientations/sizes/shapes and varying intensities to get a better distribution for calculating the sample mean. This is also highlighted in Fig. 9b-d. We observed that the intensity plays a more significant role, and varying the intensities within the perturbation pattern enables improved reconstruction outcomes. We added these observations to the manuscript.
>
> Finally, we sincerely appreciate the reviewer's efforts to enhance our manuscript by suggesting potential future directions for our work. We agree that the idea of reinforcement learning holds significant promise and we have revised our manuscript to provide greater clarity on intensity variations and outline possible future directions. We also added this suggestion to the future directions in Section 4.6.
>
> **Q4. Regarding Figure 2**
>
> **A.** We thank the reviewer for noticing the typo in Figure 2, left hand side has changed to $\mathcal{L}(\mathbf{x}^{(m)},\mathbf{x}_{PI})$.
>
> **Q5. Unrolled networks being the SOTA**
>
> **A.** We thank the reviewer for pointing this out. This is a direct quote from Hammernik et al, IEEE SPM, 2023, but we do agree with the reviewer that there has not been a more recent comparison this year. Thus, we modified the text to reflect this was the view as of a year ago.
>
> We note that there may be other impediments to using diffusion models in clinical MRI, as their output is stochastic with respect to the starting seed, as well as the longer inference times. However, we do not discuss these in the text to keep our focus.
>
> **Minor Comments**
>
> **A.** We thank the reviewer for their careful reading and noticing typos in our manuscript, we fixed these accordingly.

---

> > ### Author Response · Authors · 2024-11-23
> > **A Gentle Reminder**
> >
> > Dear Reviewer 4Fqf,
> >
> > We sincerely thank you for taking the time to review our responses and for your valuable feedback throughout this process. As the rebuttal period is nearing its conclusion, we hope that our revisions and clarifications have adequately addressed the issues you highlighted. If you find our responses satisfactory, we would be truly grateful if you could consider revisiting your initial evaluation.
> >
> > Should there be any remaining points or areas requiring further clarification, please do not hesitate to let us know. We are more than willing to make additional adjustments as needed.
> >
> > Thank you once again for your time and effort.
> >
> > Kind regards,
> >
> > Authors

---

### Official Review · Reviewer_aFQa · 2024-11-03

**Soundness:** 2
**Presentation:** 3
**Contribution:** 2
**Rating:** 3
**Confidence:** 3

**Summary:**

This manuscript proposed an unsupervised approach to improve MR DICOM image quality by introducing the loss term $l_{comp}+\lambda l_{pif}$, where $l_{comp}$ is to suppress noise and $l_{pif}$ is to keep fidelity to the input.

**Strengths:**

This manuscript is well-written, and the figures are well-drafted.

**Weaknesses:**

1. The motivation is weak. Starting reconstruction from DICOM output rather than raw k-space is unusual for MRI practitioners, as it does not allow us to construct the encoding operator, $\mathrm{E}$.

2. The title is misleading: claiming a "physics-driven" approach while dismissing the need for raw k-space data is contradictory. In MRI, k-space data represents the underlying physical imaging process.

3. The reported results for the baseline method, scoreMRI, are questionable, with performance significantly below that in the original publication.

4. The novelty and contribution of this work are limited. The author attempted to improve the image quality with the proposed loss term $l_{comp}+\lambda l_{pif}$, where $l_{comp}$ is to suppress noise and $l_{pif}$ is to keep fidelity to the input. In my eyes, this is a simple combination of existing approaches for MRI image quality enhancement.

**Questions:**

1. Refer to Weakness 1, how the coil sensitivity and sampling pattern are handled in the $\mathrm{E}$?

2. Refer to Weakness 2, how to construct the $\mathrm{E}$ for scoreMRI?

3. The proposed method resembles SSDU, yet its performance is reportedly better to CUPID. Could you clarify the reason for this?

---

> ### Author Response · Authors · 2024-11-21
> **Response to Reviewer aFQa (Part 1)**
>
> **W1 \& Q1. Motivation of CUPID and the construction of the encoding operator, $\mathbf{E}$**
>
> **A.** As the reviewer notes, construction of the encoding operator, ${\bf E}_\Omega$ requires knowledge about the undersampling pattern $\Omega$ and the coil sensitivities. The former is fully known from the scan parameters, which are stored in the DICOM header. Thus, having the DICOM enables us to completely characterize the undersampling pattern $\Omega$.
>
> On the other hand, coil sensitivities, as we discuss in our prospective study, can be estimated (and are in the prospective study) from a separate calibration scan, e.g. see the cited reference by Krueger et al, MRM, 2023. This calibration scan is low-resolution, performed separately than the accelerated scans, and thus does not substantially affect overall scan time. These are now highlighted in the revised text for your easier reference.
>
> Thus, to summarize, construction of ${\bf E}_\Omega$ is possible with only DICOM data access, as outlined above.
>
> **W2. ``...claiming a "physics-driven" approach while dismissing the need for raw k-space data is contradictory...''**
>
> **A.** We kindly disagree with the reviewer on CUPID not being a physics-driven approach. Physics-driven deep learning (PD-DL) for MRI reconstruction utilizes the forward encoding operator $\mathbf{E}$ that comes from the MR physics knowledge and incorporates it during the data fidelity (DF) unit to measure consistency to the acquired data. Note that the $\ell_2$ fidelity term in Eq. 3 (and 4b) translates to the data fidelity approach in Eq. 5, which only requires access to ${\bf E}^H{\bf y}$, i.e. the zerofilled image and not the raw k-space, as well as the forward operator, ${\bf E}$ (discussed above) for data fidelity. Thus, our approach is still physics-driven, even though it does not need raw k-space data.
>
> **W3 and Q2. ScoreMRI results and construction of ${\bf E}$ for ScoreMRI**
>
> **A.** We thank the reviewer for this valuable comment. We acknowledge there is a performance difference compared to the original publication. For our experiments, we used the pre-trained score model that is provided by the authors in their public repository*. However, we have since noticed that they deleted their pre-trained network after we downloaded it and there is no way to reach the file right now. Thus, we suspect there may be differences between the model used in the ScoreMRI paper and the pre-trained model that was released.
>
> That being said, we do not think it would be appropriate for us to train this model from scratch and implement it, as it would not be fair to the ScoreMRI authors. Based on Rewiever 4Fqf's comments, we are currently looking into some other generative models and if we cannot fix the problem with Score MRI, we will remove its results from our manuscript.
>
> For the question on how we constructed $\mathbf{E}\_\Omega$ for ScoreMRI, we constructed it the same way we did for CUPID and all other methods, as ${\bf E}_\Omega$ should be the same for each method for fairness.
>
> *https://github.com/HJ-harry/score-MRI/tree/main

---

> ### Author Response · Authors · 2024-11-21
> **Response to Reviewer aFQa (Part 2)**
>
> **W4. Novelty of CUPID**
>
> **A.** We thank the reviewer for this comment. In existing approaches, the noise suppression happens in the network (through the proximal operator of a learned regularizer), and the fidelity happens in the data fidelity unit (through consistency with acquired raw k-space data). There are data driven methods that does this in the loss function, but this is again implemented via comparison to a reference image and/or consistency with fully-sampled raw k-space data.
>
> CUPID is the first method that uses a novel loss function to enable the training from DICOM images acquired *at the target acceleration rates* (*i.e.*, clinical images that still have artifacts in them). This is enabled by a completely novel parallel imaging fidelity loss $\mathcal{L}_{pif}$ that uses information about the physics of parallel imaging in its design. Thus, we think it is not accurate to reduce our novelty and contribution to a "summation" of broad terms, since what is inside the terms is what matters.
>
> Furthermore, CUPID distinguishes itself from both existing PD-DL and data-driven methods that require raw k-space data (fully or undersampled) for training, *and* from generative methods that use reference fully-sampled DICOM images to train a likelihood or score function, further highlighting CUPID's novelty.
>
> **Q3. Difference to SSDU**
>
> **A.** We kindly disagree with the reviewer that our method resembles SSDU. SSDU partitions the available raw k-space data into 2 disjoint sets ($\Omega = \Lambda \cup \Theta)$ and trains the network on one set while defining the loss function on the other one. As one can see, SSDU still uses raw k-space data and cannot operate in scenarios where this is unavailable.
>
> We emphasize that CUPID uses *only DICOM images* acquired *at the target acceleration rates* to allow database or subject-specific training/fine-tuning even if the raw k-space data are not present. Therefore, it is expected to see a decrease in quantitative results to SOTA methods that use raw k-space data *that further benefit from redundancies from across multiple coils*. However, CUPID achieves competitive results, which is also highlighted by Reviewer 4Fqf and Reviewer snDg.

---

> > ### Author Response · Authors · 2024-11-23
> > **A Gentle Reminder**
> >
> > Dear Reviewer aFQa,
> >
> > We sincerely thank you for taking the time to review our responses and for your valuable feedback throughout this process. As the rebuttal period is nearing its conclusion, we hope that our revisions and clarifications have adequately addressed the issues you highlighted. If you find our responses satisfactory, we would be truly grateful if you could consider revisiting your initial evaluation.
> >
> > Should there be any remaining points or areas requiring further clarification, please do not hesitate to let us know. We are more than willing to make additional adjustments as needed.
> >
> > Thank you once again for your time and effort.
> >
> > Kind regards,
> >
> > Authors

---

> ### Comment · Reviewer_aFQa · 2024-11-25
>
> I would like to thank the authors for their efforts in revising the manuscript. However, I have decided to retain my original rating for the following reasons:
>
> It remains unclear whether obtaining raw data is genuinely challenging, even for equitable fast MRI, particularly since a separate scan is required to acquire coil sensitivity for reconstruction. Moreover, in this case, the performance of the proposed method falls short compared to SSDU. Given these considerations, it is difficult to identify a compelling reason to adjust the rating. Nevertheless, I sincerely appreciate the authors' efforts. If you have any questions, please feel free to reach out.

---

> > ### Author Response · Authors · 2024-11-26
> > **Official Comment by Authors**
> >
> > We thank the reviewer for taking the time to review our revised manuscript. We sincerely hope that the responses provided below effectively address the reviewer’s remaining confusion.
> >
> > **No raw data needed in the pipeline**
> >
> > We want to emphasize that **our pipeline does not require access to raw data at any stage**. The separate scan for coil sensitivity estimation only requires DICOM access, and thus operates independently of any raw data access. Moreover, CUPID is compatible with any algorithm that estimates and updates sensitivity maps alongside the output images (*e.g.*, doi: 10.1007/978-3-030-87231-1_34). However, exploring this aspect was not the primary focus of our study.
> >
> > **Why training without raw data access is important?**
> >
> > The reality is that the majority of scanners outside specialized or academic MRI centers—such as those in local hospitals or mobile MRI units—lack the ability to access or export raw k-space data due to the absence of vendor agreements. Furthermore, the use of DL-based reconstruction in clinical settings is limited by generalizability issues and artifacts. Studies attribute this to mismatches between training data and target populations, leading to poor performance on out-of-distribution test samples. This was a concern raised by Reviewer CGvy as well, and we kindly encourage the reviewer to refer to the responses we provided there as well. Succinctly, from a 2024 article, doi:10.1007/s10334-024-01173-8:
> >
> > *Good generalization of DL-based MRI reconstruction models is critical for clinical workflows. However, achieving a good generalization is challenging because MRI data can vary substantially in terms of different factors, e.g. MRI hardware, vendor-specific scanning protocols, patient populations, and the anatomical regions being imaged. This variability can lead to a model that performs well on data from one source but poorly on data from another, a phenomenon known as domain shift or distribution shift. [...]  In cases where only a few training examples from a target domain are available, pre-training a network on other data and fine-tuning it to the target domain can improve performance. In the challenging case where no target data are available for fine-tuning, test-time-training, which involves adapting to a single training at inference, is a viable performance-enhancing alternative.*
> >
> > **As a result, acquiring raw data and training DL reconstructions for specific MR acquisitions/target populations without any raw-data access remains a challenge in MRI.** This is the problem we are addressing.
> >
> > **Performance difference to SSDU, which has raw data access**
> >
> > As for the comparison to SSDU, SSDU relies on raw k-space data and is unable to function in situations where such data are unavailable. In contrast, CUPID exclusively utilizes DICOM images acquired at the target acceleration rates, enabling database or subject-specific training or fine-tuning even in the absence of raw k-space data. This naturally leads to a slight reduction in quantitative performance compared to SSDU which leverages raw k-space data and exploit redundancies across multiple coils. **A similar observation can be made when comparing SSDU to supervised learning**, where the latter outperforms SSDU, due to the fewer number of measurements available to SSDU. In CUPID, resources are further restricted, which explains the slight difference in performance.
> >
> > However, just as SSDU solves a training issue where supervised learning cannot be used (*i.e.*, when there is no reference data), CUPID solves an issue where SSDU/supervised learning cannot be used (*i.e.*, when there is only access to DICOM images).

---

### Meta-Review · Area_Chair_K1qw · 2024-12-22

**Metareview:**

This paper proposes an unsupervised MRI reconstruction method that directly operates on under-sampled DICOM images, eliminating the need for raw k-space data. The approach introduces a reweighted L1 loss for image compressibility and a consistency loss under random perturbations to suppress noise and maintain fidelity. The motivation stems from enabling reconstruction in settings where access to raw k-space data is limited, such as rural regions. Experiments on knee and brain scans from the fastMRI dataset demonstrate the method’s effectiveness with 4x acceleration and various undersampling patterns, including prospective undersampled images.

Strength: The manuscript tries to address a real-world challenge by focusing on MRI reconstruction without raw k-space data. Its less restrictive data access assumption compared to fully- and self-supervised methods could be a flexibility as claimed. The method is thoroughly evaluated in diverse settings, including both retrospective and prospective undersampling. Especially, the inclusion of experiments on prospective undersampled MRI reconstruction highlights the method's practical relevance.

Weakness: A main concern shared by most reviewers is about the paper's motivation, to start the reconstruction from DICOM output rather than raw k-space is unconventional, which may be unusual for normal MRI reconstruction setting. While the author’s reasoning could make sense especially for rural regions, as reviewer pointed out, there are already extensive raw k-space datasets and pre-trained models available publicly. Though adaptation may be needed when generalized to different scanners or patients, the raw k-space data can be utilized during testing-time adaptation. This may undermine the need for this approach, especially since all MRI scanners inherently sample raw k-space data. Another outstanding concern shared by reviewers is that the reported results for the baseline, ScoreMRI, can be questionable, with performance significantly below the original publication. Besides, more state-of-the-art unsupervised approaches based on generative models should also be discussed and compared. Moreover, the results exhibit over-smoothed textures. Finally, the presentation in the manuscript can be further improved in writing quality.

Overall, considering the important concerns not well addressed in the current version, I suggest rejection and that the paper could be improved by a major revision.

**Additional Comments On Reviewer Discussion:**

The author provides detailed response in the rebuttal, but unfortunately only half of the reviewers responded even after quite a few reminders. From the reviewers' follow-up, not all the concerns are fully addressed. After reading the paper, review comments and rebuttal, I agree some concerns still stand out as described in the weakness section above.

---

### Decision · Program_Chairs · 2025-01-22

Reject